# SELF-SUPERVISED SPEECH ENHANCEMENT USING MULTI-MODAL DATA

## ABSTRACT

Modern earphones come equipped with microphones and inertial measurement units (IMU). When a user wears the earphone, the IMU can serve as a second modality for detecting speech signals. Specifically, as humans speak to their earphones (e.g., during phone calls), the throat's vibrations propagate through the skull to ultimately induce a vibration in the IMU. The IMU data is heavily distorted (compared to the microphone's recordings), but IMUs offer a critical advantage — *they are not interfered by ambient sounds*. This presents an opportunity in multi-modal speech enhancement, i.e., can the distorted but uninterfered IMU signal enhance the user's speech when the microphone's signal suffers from strong ambient interference and mitigate the need of labeled data for model development?

We combine the best of both modalities (microphone and IMU) by designing a *cooperative and self-supervised* network architecture that does not rely on clean speech data from the user. Instead, using only noisy speech recordings, the IMU learns to give hints on where the target speech is likely located. The microphone uses this hint to enrich the speech signal, which then trains the IMU to improve subsequent hints. This iterative approach yields promising results, comparable to a supervised denoiser trained on clean speech signals. When clean signals are also available to our architecture, we observe promising SI-SNR improvement. We believe this result can aid speech-related applications in earphones and hearing aids, and potentially generalize to others, like audio-visual denoising.

## 1 INTRODUCTION

Speech enhancement/denoising are long-standing problems in audio analysis. The recent deep learning approaches have successfully broken through the performance walls, to the extent that even pre-trained voice assistants like Siri, Alexa, Google are remarkably successful (Tulsiani et al., 2020). It is not surprising that the bar on speech enhancement is getting raised, with newer form-factors and more challenging use-cases in the horizon. A growing domain of interest is in the context of "earables" (e.g., earphones, hearing aids, and glasses). Even though the user speaks from close to the earphone, the problem is particularly challenging because: (1) the background interference can be high in real-world public environments (e.g., restaurants, airports, busses, trains) (Schwartz, 2022). (2) Users tend to speak softly, lest they disturb others around them. Finally, (3) the relatively fewer microphones on earable devices must forgo some of the array processing gains (compared to, say, table-top devices such as Amazon Alexa or teleconferencing systems). In sum, the SINR (Signal to Interference Noise Ratio) of the target speech signal can be very low in real-world scenarios.

Although challenging, unique opportunities emerge as well. Modern earphones are equipped with IMUs that sense the vibrations due to human speech (Jabra, 2022). Of course, the IMU's sampling rate is $\approx 400\,Hz$, hence the recording of the human speech is heavily aliased and distorted by the non-linear human bone-channel (Blue et al., 2013). However, ambient sounds do not induce vibrations in the IMU, implying that the *IMU signal remains immune to external interference*. The microphone on the other hand records a high quality signal from the user's mouth ($44\,kHz$), but can be heavily polluted by ambient interference. Non-stationary interference, such as voices of other people, are difficult to denoise; even today's best denoisers (Wang & Chen, 2018), that perform remarkably well on stationary noise or on pre-trained distributions, falter against speech and music. Moreover, existing techniques mostly require clean speech for training the models. With multi-modal

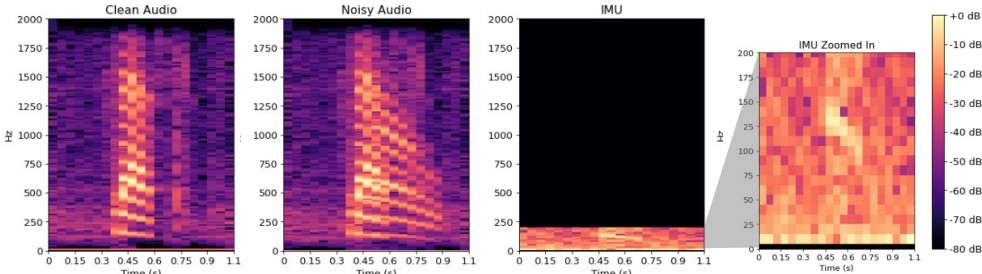

Figure 1: Earphone measurements: (a) Audio signal recorded without interference, (b) Audio recording with interference, (c) IMU recording from earphone. (d) Zoomed in view of IMU signal.

data from both the microphone and IMU (see spectrograms in Fig. 1), we see an opportunity to close gaps in speech enhancement and remove the necessity of collecting clean and labeled speech.

We propose a self-supervised architecture that does not rely on clean speech data to train the network. Instead, we utilize the everyday, noisy recordings from the earphone that the user can record on the fly. The key idea is to develop a cooperation between the IMU and the microphone, so each modality can teach and learn from the other. To this end, our architecture, called IMU-to-Voice (*IMUV*), is composed of two separate models — a `Translator` and a `Denoiser`.

Briefly, the Translator translates the distorted IMU signal to higher-resolution audio, and then constructs a time-frequency mask to crudely identify the locations of user's speech. The Denoiser, which only has noisy speech signals, uses this crude mask to slightly enrich the user's speech signal. The Denoiser's output — the slightly enriched speech signal — now offers a reference to the Translator to learn a better mask, which is in turn fed back to the Denoiser to further enrich the speech signal. The iterations converge to an SNR-enhanced voice signal at the output of the Denoiser, even in the presence of strong interference. Importantly, no clean speech is needed to bootstrap or train this network; the noisy voice signal can even be at $0$ dB SINR.

Zooming out to a higher level, the results are not surprising. Given that the IMU is completely unaffected by strong interference, it should be able to guide the audio Denoiser down the correct path of gradient descent. The only risk emerges from the fact that the IMU has no way to validate whether its guidance is correct, and given the IMU is heavily distorted, it is easy to make mistakes. However, this is where we find that even a noisy voice recording gives the needed validation to the IMU, so the Translator and Denoiser can teach each other and independently descend in the right direction.

We show that our proposed two-step model actually inherits the structure of *expectation maximization* (EM) (**?**), with the likelihood and posterior functions estimated by neural networks. EM is known to be sensitive to its initial condition, similar to how the initial mask from the Translator is crucial for downstream convergence. Although our Translator is able to provide one acceptable mask, the question around the optimal mask (upper bound), or the minimally adequate mask to ensure convergence (lower bound), remains open. We leave this to future research.

**Summary of Results:** With help from $4$ volunteers, we gathered IMU and microphone data from earphones, and injected interference from a public audio dataset into the microphone data stream. The *self-supervised* IMUV model is trained on this unclean dataset (at varying SINR levels). We evaluate the final denoised signal using two metrics: scale invariant signal to noise ratio (SI-SNR), and word error rate (WER) from an automatic speech recognizer (ASR) (Yu & Deng, 2016). Results show that in terms of WER, *self-supervised* IMUV is comparable with the *supervised audio Denoiser* (trained with clean voice data), achieving less than $1\%$ difference. When we allow *IMUV* to also train on clean signals, *supervised* IMUV exceeds *self-supervised* IMUV by $5\%$. Finally, when using SI-SNR as the metric, the gains are higher and more consistent with both *supervised* and *self-supervised* IMUV. In closing, we find that IMU extends only one of two advantages — we can either choose to improve denoising performance, or relieve the user from collecting clean voice data.

## 2 MULTI-MODAL SELF-SUPERVISION

### 2.1 PROBLEM STATEMENT

We consider two input streams: a high-resolution audio signal $H$ from the microphone, and a low-resolution surface-vibration signal $L$ from the IMU. Since all recordings are from everyday

environments, $H$ is composed of three parts: the speech signal from the target user, $H_T$, the interfering signals from nearby people, $H_I$, and the hardware/ambient noises, $H_N$. Thus, $H = H_T + H_I + H_N$. We assume no knowledge of $H_T$, $H_I$, or $H_N$.

The IMU signal $L$ consists of only two parts: the target vibration from the user, $L_T$, and the hardware noise, $L_N$. We assume no knowledge of either $L_T$ or $L_N$. Also, since the vibrations are essentially an outcome of the user's speech, $L_T$ is a non-linear projection of $H_T$, i.e., $L_T = f(H_T)$. This projection is expected to be different across users, depending on each user's bone, muscle, and tissue conduction properties.

The final output of our model is expected to be a denoised high-resolution audio signal $\hat{H}_T$, containing only the speech of the target user, $T$.

## 2.2 NETWORK ARCHITECTURE

**Translator design**: Figure 2 shows the proposed network architecture, with a *Translator* on top and a *Denoiser* below it. For self-supervision, the Translator needs to supply a reference signal to the Denoiser. This means the vibration signal $L$ at $400\ Hz$ needs to be translated to an audio signal $H_T'$ that approximates the target speech signal. Since $H_T'$ needs to be at, say, $16\ kHz$, the Translator's task is that of super-resolution. This large up-sampling factor from $400\ Hz$ to $16\ KHz$ is prone to overfitting with a conventional auto-encoder. Hence, we design the network as a guided autoencoder to inherit earlier successes in (Lai et al., 2017). The idea is to up-sample the signal in multiple stages, each stage with a small up-scaling factor and a corresponding stage loss. Using a 3-stage decoder, we up-sample $L$, represented in the time-frequency (TF) domain, from $400\ Hz$ to 800, 3200, and finally to $16\ KHz$. The final loss is regularized by the individual stage losses to curb the decoder from overfitting. Of course, the reference signal for computing loss is the noisy audio signal $H$ from the microphone (but in subsequent rounds, becomes the output of the Denoiser). Though we train the translator with noisy audio as ground truth, we observed that the translated audio (mask) does not contain high energy for noises. As the noises are absent in the IMU data and are random, i.e, they have different sources and are at various places for different samples, the translator can not learn the noise signals. On the other hand, the translator can identify the target audio as its aliased version is present in the IMU data.

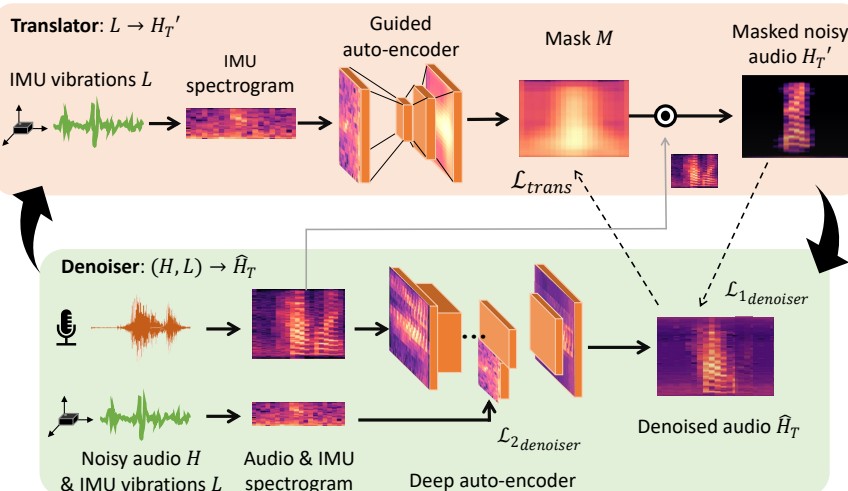

Figure 2: Proposed architecture composed of a Translator on top, that attempts super-resolution from $L$ to $H$, and generates a mask. The model below is a multimodal Denoiser that uses the mask to enhance the target audio. The enhanced target audio is returned to the Translator to improve the super-resolution, resulting in an improved mask. This iteration continues.

**Mask generation**: The heavy up-sampling, and that too with a noisy reference signal, cannot be expected to yield fine-grained speech (with accurate amplitude and phase in each time-frequency (TF) bin). Instead, the spectrogram can only be expected to have marked TF bins where the target audio dominates. Hence, we treat the output spectrogram of the guided auto-encoder as an *mask* (Vaswani

et al., 2017) for $H_T$ — this map $M$ gives us $p(H_T | \hat{H}_T)$ for each TF bin. To derive an approximate audio signal, $H_T'$, we pass the mask through a *Sigmoid* function to first generate a mask, and then multiply the mask element-wise with the denoiser's output $\hat{H}_T$ to obtain $H_T' = \hat{H}_T \odot \sigma(M)$. Of course, in the initial round of mask generation, $\hat{H}_T = H$.

**Denoiser design**: The Denoiser's input is both $H$ and $L$ and the output should be the denoised signal $\hat{H}_T$. One option is to train a network to learn the end-to-end mapping from the input data $(H, L)$ to the desired output $\hat{H}_T$. However, without clean audio data, this mapping does not converge well, or is slow and data hungry. More importantly, we know that a consistent mapping exists between audio and IMU, dictated by the bone channel that conducts the throat's vibration. Our network architecture must incorporate this knowledge.

We design an auto-encoder (AE) using only the audio $H$ as input, however, we force part of the latent space to match the IMU signal $L$. Specifically, we design the AE's latent space as $Z = \{Z_L, Z_H\}$ (see Fig. 3) and force $Z_L$ to match the IMU data $L$ (we detail the loss terms in the next section). Since the IMU only contains low frequencies ($\leq 400\ Hz$) and contains no interference, we intend the remaining latent space $Z_H = Z \setminus Z_L$ to capture the gap between audio and IMU. Hence, we model $Z_H = \{Z_T^{(hi)}, Z_I^{(all)}\}$, where $Z_T^{(hi)}$ is a representation of the target's *high* frequency components, and $Z_I^{(all)}$ is a very compressed representation of *all* the interfering signals.

Assuming the interference is uncorrelated to the target user's speech, we add a loss term between $Z_I^{(all)}$ and the IMU signal $L$ to enforce contrast between them. We also add another loss term between $Z_T^{(hi)}$ and $L$ to enforce their correlation. Finally, the decoder uses only $\{Z_L, Z_T^{(hi)}\}$ to reconstruct the denoised audio signal, $\hat{H}_T$, and trains it against the Translator's output, $H_T'$.

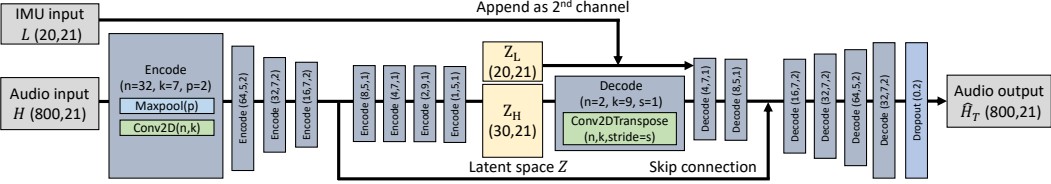

Figure 3: Denoiser architecture: The audio is encoded into a latent space, one part of which mimics the IMU and the other parts are representations of high frequency speech signals and interference.

The Denoiser is almost ready, except for one small detail. To utilize the IMU data $L$ during test time as well, we concatenate $L$ as a second channel, alongside $Z_L$. The 2 channels serve as the first layer of the decoder. To match the dimensions, $Z_H$ progresses through one additional layer of decoding. Although we design $Z_L$ to match $L$, it's important to concatenate $L$ because it does not contain any ambient noises. Since $Z_L$ and $L$ both represents the low-frequency target signal, subsequent layers will learn the weights from both modalities.

**Training**: The Translator begins by training against the noisy audio $H$. After $N_t = 25$ epochs, we freeze the Translator and use its output (i.e., the masked audio $H_T'$) to train the Denoiser for the next $N_d = 75$ epochs. We denote $(N_t + N_d)$ epochs as a training cycle. We then start the next cycle by freezing the Denoiser and using the denoised signal $\hat{H}_T$ *from the previous cycle* to train the Translator. The iteration is performed for $C = 3$ cycles.

Fig. 4 shows snapshots from the start and end of the training process. The first column in Fig. 4 plots the spectrogram of clean target speech $H_T$ on top, and the interfered audio $H$ at the bottom. The top of the second column shows the Translator's mask after the first training cycle; evidently, IMU offers a crude map $M$ at this time. The bottom of the second column plots the Denoiser's output when it has been trained using the masked audio, $H_T' = H \odot \sigma(M)$. The top of the third column shows the mask after the last training cycle – the improvement is visible. We observe that the Translator converges reasonably well because the interference varies over time, preventing the Translator from overfitting to the interference. Finally, the bottom of column 3 shows the denoised audio $\hat{H}_T$ using the final mask; this is our final output.

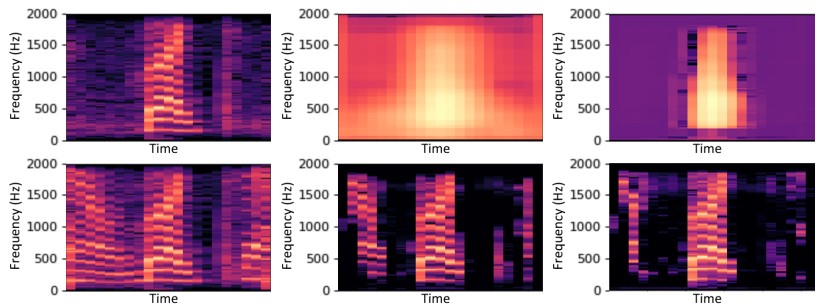 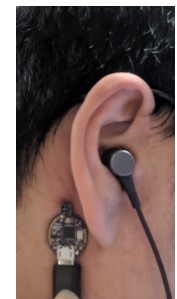

Figure 4: (Column 1) Spectrogram of clean target signal $H_T$ on top, and the noisy microphone signal $H$ at the bottom. (Column 2) Translator's mask after the first cycle on top, and the Denoiser's output after the first cycle at the bottom. (Column 3) Translator's mask after the *last* cycle on top, and the Denoiser's output after the *last* cycle at the bottom.

Figure 5: IMU placement for data collection

## 2.3   LOSS FUNCTIONS

**Translator's loss function**: Aggressive up-sampling is prone to overfitting, so the Translator incorporates a loss function at each stage of the guided auto-encoder. The final loss is a convex combination of Mean Absolute Error (MAE):

$$\mathcal{L}_{trans} = \mathbb{E}_{x \sim p(x)} \frac{\sum_{i=1}^{n} w_i ||D_{-1}(x)_i - T(x)_i||_1}{\sum_{i=1}^{n} w_i} \tag{1}$$

where $n$ is the number of scale-up stages; $w_i$ is the weight for stage $i$; $D_{-1}(x)_i$ is the Denoiser's output, down-sampled to match stage $i$; and $T(x)_i$ is the Translator's output after stage $i$.

**The Denoiser's loss function** is composed of three terms as follows:

$$\mathcal{L}_{denoiser} = \mathcal{L}_H + \lambda_1 * \mathcal{L}_L + \lambda_2 * \mathcal{L}_C \tag{2}$$

where $\mathcal{L}_H$ denotes the *audio reconstruction loss*; $\mathcal{L}_L$ is the *IMU loss* from the latent space; $\mathcal{L}_C$ is the *correlation loss*, and $\lambda$ is the weighing scalar. The loss functions are defined as:

$$\mathcal{L}_H = \mathbb{E}_{x \sim p(x)} ||T(x) - D(x)||_1 \qquad \mathcal{L}_L = \mathbb{E}_{x \sim p(x)} ||L - Z_L||_1 \tag{3}$$

$$\mathcal{L}_C = \mathbb{E}_{x \sim p(x)} \sum_{i,j} abs(corrcoef(L_i, Z_{I,j}^{(all)})) - \sum_{i,k} abs(corrcoef(L_i, Z_{T,k}^{(hi)})) \tag{4}$$

The *Correlation loss* $\mathcal{L}_C$ aims to capture the uncorrelated relationship between the IMU signal $L$ and the interference $Z_I^{(all)}$, as well as the correlation between the IMU $L$ and the high frequency components of the speech, $Z_T^{(hi)}$. The negative sign for the second term indicates that higher correlation reduces the loss function (and vice versa for the first term). In the equation, $i, j, k$ are the indices of the dimensions of $L$, $Z_I^{(all)}$, and $Z_T^{(hi)}$. We calculate the absolute value of correlation coefficients to account for harmonic behaviors in the speech signals. Algorithm 1 shows the pseudo-code for training self-supervised *IMUV*.

## 2.4   ITERATIVE BEHAVIOR

Our iterative training process inherently mimics the Expectation-Maximization framework (Dempster et al., 1977). For ease of explanation, consider $L$ and $H$ denoting the spectrogram of IMU and audio (instead of the time domain signals). Observe that the Translator poses as the first E-step; it calculates the distribution of the latent variable, which is the mask $M = \{m_i\}$ for each TF bin $i$. Given the low-resolution vibration data $L$ as well as $\hat{H}_T$ from the Denoiser, it outputs $p(\{m_i\}|L, \hat{H}_T)$, where $m_i = p(H_T|H)_i$ is the mask of the target signal. By injecting the denoiser loss into the distribution, we get the E-step function:

$$Q_{\hat{H}_{T,new}|\hat{H}_T} = \mathbb{E}_{M|L,\hat{H}_T}[\mathcal{L}_{denoiser}]$$
$$= \mathbb{E}_{M|L,\hat{H}_T}[F(\hat{H}_{T,new}; \hat{H}_T, M)]$$

---

**Algorithm 1** Self-supervised *IMUV* Training ($H$,$L$, $Translator$, $Denoiser$)

---

1: $\hat{H}_T \leftarrow H$
2: **for** $c \leftarrow 0$ to $C$ **do**
3:     $Translator.loss \leftarrow \mathcal{L}_{trans}(w, \hat{H}_T)$
4:     $Translator.train(input \leftarrow L, epoch \leftarrow N_t)$
5:     $M \leftarrow Translator.predict(L)$
6:     $H_T{}' \leftarrow H \odot \sigma(M)$
7:     *// Freeze the translator and train the denoiser*
8:     $Denoiser.loss \leftarrow \mathcal{L}_{denoiser}(H_T{}')$
9:     $Denoiser.train(input \leftarrow (H, L), epoch \leftarrow N_d)$
10:     $\hat{H}_T \leftarrow Denoiser.predict(H, L)$
11:     *// Freeze denoiser and train translator in next cycle*
12: **end for**

---

where $\hat{H}_{T,new}$ is the Denoiser's output in the new cycle, and $F(\hat{H}_{T,new}; \hat{H}_T, M) = ||\hat{H}_{T,new} - \hat{H}_T \odot M||_1$. Lastly, the Denoiser training can be viewed as the M-step, where we find the best target signal estimation $\hat{H}_{T,new}$ minimizing the expected loss $Q$.

## 3 EXPERIMENTS AND RESULTS

### 3.1 EXPERIMENT SETUP:

**Dataset Construction.** We recruit 4 volunteers and ask them to wear a normal earphone and a separate IMU (Fem, 2022) near their ears – Figure 5 shows the set-up. A separate IMU is needed since today's earphones do not make the recorded IMU data accessible. Each volunteer speaks 39 different keywords prescribed by the Google's Speech Command dataset (Warden, 2018), as well as wakewords like Google, Siri, Bixby, and Alexa – each word is repeated 10 times. The measurements are performed in a quiet room, and serve as $H_T$. The earphone's microphone samples the audio at $44.1\,KHz$ and we sub-sample to $4\,KHz$ to mimic phone calls over earphones (Lee et al., 2016). The IMU is sampled at $400\,Hz$. *We have currently published the dataset on GitHub anonymously* (IMU, 2022b). To the best of our knowledge, this is the first speech dataset composed of synchronized audio and IMU vibrations from the ear-location.

To synthesize noisy signals $H$, we randomly draw audio samples from Google's speech command dataset (Warden, 2018), containing voices of $2,618$ human speakers. These samples serve as $H_I$. Unless specified otherwise, we synthesize $H$ at 5 dB SIR. The IMU signals, on the other hand, need no synthesis, so we automatically have $L$. The total dataset $\langle H, L \rangle$ is now ready and extends over 990 hours.

**Performance Metrics.**

(i) We use *scale invariant signal to noise ratio* (SI-SNR) as the main evaluation metric (Le Roux et al., 2019). SI-SNR is computed as the ratio between the correlation of our output signal with the target signal, as follows:

$$\text{SI-SNR} = 10log_{10}\frac{||\frac{\hat{H}_T^* H_T}{||H_T||^2} H_T||^2}{||\frac{\hat{H}_T^* H_T}{||H_T||^2} H_T - \hat{H}_T^*||^2}.$$

(ii) We also report the *word recognition accuracy* of the denoised signal, using Google's Key Word Spotting Classifier (KWS) with 10 and 35 classes, denoted as `KWS10` and `KWS35`, respectively (Rybakov et al., 2020; Goo, 2022). We re-trained the KWS models using $4\,KHz$ audio data.

**Models for Comparison.**
*(1) Unprocessed:*          The raw audio without denoising
*(2) Supervised Denoiser:*  A recent speech enhancement model (Park & Lee, 2016)
                            trained on clean speech; 216K parameters.
*(3) Supervised* IMUV*:*    Our proposed model trained on clean speech; 60K parameters.
*(4) Self-Supervised* IMUV*:* Our proposed iterative model in Figure 2; 180K parameters.

For fairness, we train all the above models using the same dataset. We employ a "no overlapped words between training and testing set" policy. Around $16\%$ of the dataset is set aside for testing and the rest for training. We also ensure that $H_T$ and $H_I$ are different words. We publish the codes in the GitHub Repository (IMU, 2022b), and denoised audio samples on the GitHub site (IMU, 2022a).

## 3.2 RESULTS

**Overall performance:** Table 1 reports comparative results across all metrics and models. Unsurprisingly, *supervised IMUV* substantively outperforms all models across all metrics. On the other hand, *self-supervised IMUV* is comparable to *supervised Denoiser*, outperforming marginally in SI-SNR but is weaker in KWS metrics. This distills the contribution of *IMUV* to speech enhancement as follows: we can either choose to obtain $0.6 - 1$ (personal) to $0.2 - 3.1$ (general) SI-SNR gain while requiring the user to provide clean speech data or relieve the user from the data collection burden at the cost of sacrificing that same performance gain. Both are not possible yet.

Table 1: Performance comparison across models and metrics.

| Models | Personal Model | | | General Model | | |
| --- | --- | --- | --- | --- | --- | --- |
| | SI-SNR (dB) | Accuracy(%) KWS10 | Acc.(%) KWS35 | SI-SNR (dB) | Acc.(%) KWS10 | Acc.(%) KWS35 |
| Unprocessed | 5.3 | 49.24 | 48.92 | 5.3 | 49.24 | 48.92 |
| Supervised Denoiser | 8.3 | 65.72 | 64.02 | 2.9 | 51.37 | 46.22 |
| Supervised *IMUV* | 9.9 | 69.02 | 66.78 | 6.0 | 55.27 | 52.96 |
| Self-supervised *IMUV* | 9.3 | 64.45 | 62.83 | 5.8 | 48.67 | 45.19 |

**Personalized versus generalized models:** Table 1 compares the accuracy of a personalized model (where training data is drawn from the target user) and a generalized model (where the training data is drawn from users other than the target user). Understandably, the accuracy improves across all models when the training data is personalized, however, the *self-supervised IMUV* performs slightly better than the *supervised Denoiser* for generalized models. This implies that the marginal gain from the IMU is higher in generalized models, since with personalized models, the *supervised Denoiser* may be able to learn user-specific voice patterns (e.g., base frequency, pitch, prosody, harmonics, etc.) Observe that the generalized models are valuable during the phase when on-the-fly data is being gathered for self-supervision. In the case of earphones, for example, the user would buy an earphone with the pre-installed generalized model and personalize it over time by training with recorded data from everyday life.

**Variation across users:** Figure 6 plots both the SI-SNR and KWS accuracy across 4 users. *supervised IMUV* consistently benefits from both the IMU and the clean-data supervision, and as expected, the *supervised Denoiser* and *self-supervised IMUV* are mostly comparable. With users who tend to produce high frequency voices/sounds, the gain from IMU is relatively less since the IMU does not capture these frequencies; the *supervised Denoiser* avails an advantage for these voice characteristics.

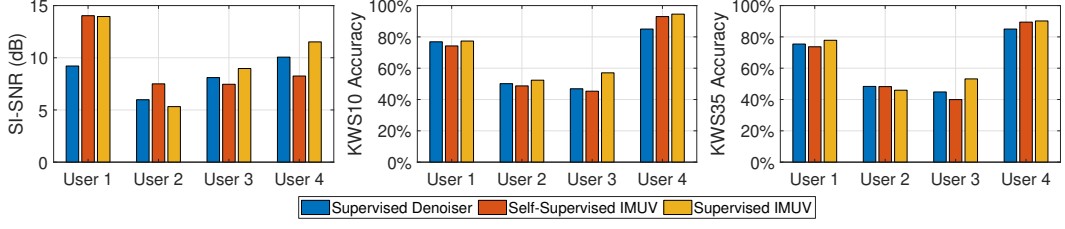

Figure 6: Performance across different users for (a) SI-SNR, (b) KWS 10, (c) KWS 35 accuracy.

**Varying mix of clean and interfered data:** *self-supervised IMUV* has trained entirely on interfered data. Figure 7 shows the effect of some fraction of the training data being clean; the testing data also follows the same training data distribution. This is likely to be the average case in our earphone application, where the user may sometimes speak in a silent environment. In such settings, the gain of *self-supervised* IMUV over *supervised Denoiser* is not affected by the fraction of clean data.

**Varying interference:** Figure 8 plots SI-SNR against varying SIR (Signal to Interference Ratio) of the training/testing data. The contribution of IMU grows as the SIR drops since the additional IMU modality becomes more valuable under more noisy environments. This explains why *self-supervised IMUV* outperforms *supervised Denoiser* at low SIRs, but is worse at higher SNRs where the gain from IMU's guidance is offset by the penalty of self-supervision.

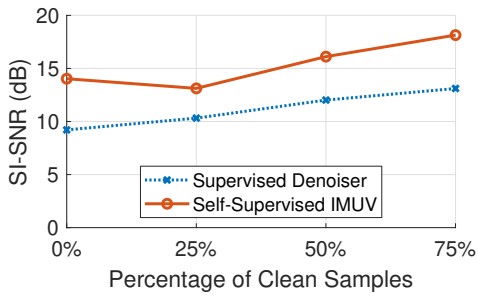

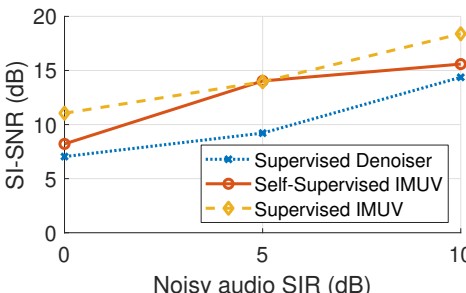

Figure 7: *IMUV* offers gain with increasing percentage of clean-data in training samples.

Figure 8: Performance versus SIR regimes.

**Effects of the skip connection in the Denoiser Architecture:** Although our network is configured by several hyper-parameters, we report the performance variation against one of them — the weights assigned to skip connections. In a conventional auto-encoder (AE), skip connections are used to mitigate vanishing gradients in the deep network (Tong et al., 2017). However, in this work, we are injecting the IMU input $L$ in the latent space, which will be "skipped" by the skip connections. Hence, we analyze the effect of regulating the skip connection in *IMUV*. Table 2 compares the performance for different skip connection weights in the Denoiser network. Since the skip connection carries some fine-grained speech patterns (that may not be captured in the latent space), the SI-SNR drops if we remove the skip connection (0%). Here, 0% skip means that we do not change the hierarchy but just change the weights of the skip connections to 0%, when it joins the latent space. On the other hand, increasing the skip connection weights (100%) reduces the gain from clean IMU in the latent space. Consequently, we use a moderate skip connection weight in *IMUV*.

Table 2: Performance for varying weights on the skip connection in the Denoiser. The metrics are [SI-SNR(dB) / KWS 10 accuracy(%) / KWS 35 accuracy(%)]. $P\%$ denotes the skip connection contributes $P\%$ weight when joining the decoder latent space. $100\%$ setting ignores all inputs from the latent space, and only takes the results from the skip connection.

|  | 0% | 10% | 50% | 100% |
|---|---|---|---|---|
| Self-S. *IMUV* | 11.0 / 68.4 / 70 | 13.9 / 74.2 / 73.7 | 12.1 / 65.3 / 68.6 | 11.1 / 65.1 / 70.1 |
| Super. *IMUV* | 12.1 / 80.2 / 79.8 | 14 / 81.4 / 77.8 | 12.5 / 82.5 / 82.16 | 13.6 / 83.0 / 83.0 |

# 4 RELATED WORK

**Multi-modal speech enhancement:** The closest work to this paper is SEANet (Tagliasacchi et al., 2020) that uses both audio and IMU through a wave-to-wave convolutional generator and discriminator architecture. The core architecture builds on (Kumar et al., 2019) and achieves promising *SI-SDRi* (Scale Invariant Signal to Distortion Ratio Improvement). SEANet uses accelerometer data which is unaffected by ambient noise to partially reconstruct user speech. The key difference between SEANet and our work is that, SEANet assumes the availability of clean speech reference which places a considerable onus on the user to acquire which is impractical. Our work leverages the self-supervision provided by the IMU to create the clean reference without additional burden on the user.

Another work that is closely related to ours is (Wang et al., 2021). This work uses "alias unfolding" to reconstruct user speech from low resolution IMU motion signals. This work is not multi-modal as the core idea of this work is reconstruction of spoken words from IMU signal. This is different from our work which uses IMU to denoise noisy audio. However, one commonality is that, our translator does the job of "anti-aliasing" to upsample a low resolution signal (IMU) to a high resolution signal (speech) which is similar to "alias unfolding" of (Wang et al., 2021).

**Multi-modality learning:** In recent years, many ideas have used synchronized audio and visual modalities(Hou et al., 2018; Gabbay et al., 2018; Ephrat et al., 2018; Afouras et al., 2018; Lu et al., 2019) for speech related applications (e.g., enhancement and source separation). IMUV targets a different scenario (e.g., outdoor, mobile) where a camera is unavailable to record the user's face/lips.

**Self-supervision:** Several work (Chen et al., 2021; Kashyap et al., 2021) have incorporated self-supervision in audio processing. Authors in (Wang et al., 2020) learn a latent representation of a limited set of clean speech and uses noisy speech to share a latent representation with the clean examples, reducing the burden of clean data. This bears similarity to IMUV, however, we face the challenge of not having initial clean information to descend down the correct gradient. The work done by (Kashyap et al., 2021) uses a 20 layered Deep Complex U-Net to perform self-supervised speech denoising with noisy speech targets. The authors leverage two key conditions that the noise distribution must adhere to: (1) The input and target noises are sampled from zero-mean distributions and are uncorrelated to the speech input. (2) The correlation between the noise in the input and in the target is close to zero. In contrast, in IMUV we make no such assumptions about the noise distribution as the availability of another modality (IMU) provides more information.

**Input Representation:** The performance of any denoiser is impacted by the representation of the speech signal. Authors in (Nossier et al., 2020) compare the impact of TF representations on noise reduction. Most deep learning approaches can be broadly classified based on whether they operate on raw audio waveforms (Kolbæk et al., 2020; Pandey & Wang, 2019) and WaveNets for raw audio waveforms (Germain et al., 2018), STFT spectrograms (Kumar & Florencio, 2016; Lu et al., 2013), or other audio representations like Mel Frequency Cepstral Coefficients (MFCC) (Pirhosseinloo & Brumberg, 2018). Our work uses STFT spectrograms as one possible compact feature representation.

## 5 Limitations, Generalization, and Conclusion

**Mutual information between multiple-modes:** The IMU signal $L$ is a projection of the audio signal $H$ onto a lower dimensional space. This paper shows a very specific instance in which the projected (low-dimensional) signal can teach/optimize the higher dimensional signal without any supervision. The generalization of this question is of interest, i.e., how low does the lower dimensional projection need to be before (self-supervised) convergence fails. Perhaps mutual information $I(L; H)$ could shed light on this line of questioning, leading to a possible notion around the "capacity of self-supervision". We leave this to future work.

**Alternating epochs between `Translator and Denoiser`:** We have empirically chosen the number of epochs for the `Translator` and `Denoiser` as they alternate in a cycle. Although rigorously designing hyperparameters is difficult, we believe there is room to make this decision tighter, or adaptive. One idea is to model them in the explore–exploit framework, where running (the Translator or Denoiser) for longer epochs is the exploration phase, and switching is exploitation. We leave this to future work as well.

**Beyond Audio and IMU:** We envision generalization of *IMUV* in *audio-visual* speech enhancement as well. Consider a laptop camera visually recording the user's lip movements (e.g., in a video call), and the microphone recording her voice signal, polluted by ambient interference. Observe that the lip movement is akin to IMU in *IMUV*, where it is uninterfered and a low-dimensional projection of actual speech. We believe our self-supervised architecture is relevant here, offering the possibility to relax the need for clean audio data.

**Multi-modal signal synthesis:** This paper has aimed to enhance the target speech signals where the signal exists (only interference has polluted them). In other applications, the target signal may be absent (e.g., clouds completely occluding a satellite image, or a night vision camera unable to see some parts of the environment). The guidance from the second, low-dimensional, modality (e.g., RF radar that penetrates through the cloud) would help the denoiser learn the latent distribution of the target object. This learned distribution could be harnessed to synthesize/reconstruct the occluded parts, perhaps by training a GAN on top of the Denoiser.

**Conclusion**: This paper presents *IMUV*, a self-supervised speech denoising network that leverages a low resolution (IMU) modality. With an iterative network design, we show that the low-modality signal effectively guides the multi-modal denoiser. Results are promising, and could aid wearable-audio applications in the near term, and other multi-modal mixture-models in the future.

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
