# OpenReview forum: "Self-supervised Speech Enhancement using Multi-Modal Data"
_ICLR.cc/2023/Conference — Submitted to ICLR 2023_

### Official Review · Reviewer_uCfP · 2022-10-24

**Confidence:** 4
**Correctness:** 2
**Technical Novelty And Significance:** 3
**Empirical Novelty And Significance:** 3
**Recommendation:** 3

**Clarity, Quality, Novelty And Reproducibility:**

The paper is relatively easy to read, and the proposed self-supervised training for mic+accel speech enhancement seems to be novel enough. However, there is room for improvements to improve quality of the material and clarity of presentation.

Some remarks:
- Baseline "Supervised Denoiser" seems to be a relatively simple system. It's not clear how it compares to the proposed system in terms of parameters, nor how does it perform to more recent end-to-end speech enhancement work.
- Choice of sampling frequency (4kHz) is pretty odd. This sampling frequency is too low for modern communication devices. Could this design choice be clarified?
- Terminology is sometimes odd. For example, "unclean" dataset should likely be replaced with "noisy" or "perturbed". Also, "translated to an audio signal" is an unusual phrase. The whole "Translator" block could be better described as a converter, or even more generally, a mask estimator.
- "Attention map" is actually just a real-valued gain and it's basically the way speech enhancement has been done for decades
- Input and output is not well defined. It's mentioned that it's an STFT, but it's not clear if it's power, magnitude, what's the STFT setup, and if any other transformation has been applied to it
- Section 3.2: “the testing data also follows the same training data distribution”. This makes Figure 7 relatively useless, since the average SI-SNR of the test data is expected to increase, i.e., denoising is becoming easier, so increasing SI-SNR is expected.
- Reproducibility is definitely helped by the published GitHub repo. However, the paper is not clear enough on the parameter of the network and the training procedure.
- Section 5 is pitching the presented system as a solution to many problems, e.g., audio-visual enhancement and multi-modal synthesis. This seems a bit too general, especially the latter.

**Strength And Weaknesses:**

The main strength of the paper is the self-supervised training formulation, since it does not rely on clean speech signal data.
One of the main weaknesses of this paper is that it does not mention a lot of relevant prior work.
More specifically, the authors cite the following relevant work:
- SEANet: A Multi-modal Speech Enhancement Network, 2020
However, this topic has been covered extensively in more classical speech enhancement work, e.g.,
- Speech enhancement for an in‐ear communication system using bone‐conducted and acoustic signals, JASA 2001
- Speech enhancement for an in-ear communication system using optimal-filtered accelerometer signals, Acoustics 2002
- Survey of speech enhancement supported by a bone conduction microphone, ITG Speech communication 2012
- Time-Domain Multi-Modal Bone/Air Conducted Speech Enhancement, 2020
any many references in the above-mentioned.

Furthermore, even the single mentioned multi-modal speech enhancement method (SEANet) is not described correctly. More specifically, the paper claims that "uses an IMU that samples at 4 kHz. In contrast, IMUV is self-supervised [...] and operates entirely on 400 Hz, the viable sampling rate for real-world earphones." This is not exactly true. Figure 4 in the SEANet paper depicts the performance in speech and noise for accelerometer sampling rates as low as 160Hz. Moreover, those results indicate that sample rates as low as 400Hz are not severely impacting the performance of the system.

**Summary Of The Paper:**

The paper presents a self-supervised speech enhancement method for a scenario when both a microphone recording and a signal from an accelerometer sensor is available. Activity of the latter signal is well correlated with the speech signal activity, while it's also not corrupted by the acoustic noise from the environment, e.g., by interfering speech and noise. This fact is often exploited by in speech enhancement for improving the speech signal estimate.

**Summary Of The Review:**

The paper is interesting and self-supervised training is of interest for many applications.
However, the paper is not covering the existing work well enough, it's not clear how it performs to state-of-the-art speech enhancement systems, and some aspects of the design are not clear.
If the above would be addressed, I believe the paper could be considered for publication. However, in its current form, it's just not good enough from the perspective of this reviewer.

---

> ### Author Response · Authors · 2022-11-19
> **Author response to reviewer uCfP**
>
> We thank the reviewer for all the insightful and constructive comments. We address some of the
> concerns below.
>
> *Comment 1:* The comparison with multi-modal speech enhancement method (SEANet) is not described correctly.
>
> ***Response 1***: We have updated the description of SEANet. SEANet focuses on using IMU to enhance the performance, while IMUV further utilizes this IMU modality for self-supervision.
>
> *Comment 2:* Baseline "Supervised Denoiser" seems to be a relatively simple system. It's not clear how it compares to the proposed system in terms of parameters, nor how does it perform to more recent end-to-end speech enhancement work.
>
> ***Response 2***: We are reporting the number of parameters for all baseline models and proposed model.
>
> supervised denoiser: 216K, supervised IMUV: 60K, self-supervised IMUV: 180K. We have updated these numbers in Sec. 3.1.
>
> Because we are targeting the earphone application, we select a flexible learning model that can converge quickly on smaller personal dataset with speech as the interference. Recent audio-only works such as pseudo-SE [1] or MixIT [2] requires training on large public audio dataset.
>
> [1] Sivaraman, A., Kim, S., & Kim, M. (2021). Personalized speech enhancement through self-supervised data augmentation and purification. *arXiv preprint arXiv:2104.02018*
> [2] Wisdom, S., Tzinis, E., Erdogan, H., Weiss, R., Wilson, K., & Hershey, J. (2020). Unsupervised sound separation using mixture invariant training. *Advances in Neural Information Processing Systems*, *33*, 3846-3857.
>
> *Comment 3:* Choice of sampling frequency (4kHz) is pretty odd. This sampling frequency is too low for modern communication devices. Could this design choice be clarified?
>
> ***Response 3***: We are targeting the mobile and earphone applications, which need to optimize for power and computational overhead. In the speech-as-the-interference scenario, denoiser requires large networks and large amount of data to separate the personal speech pattern from the speech interference. If we operate on the high sampling rate like 44K, the amount of personal data from the user will be prohibitively large.
>
> *Comment 4:* Attention map" is actually just a real-valued gain and it's basically the way speech enhancement has been done for decades.
>
> *********************************Response 4:*********************************  We have renamed attention map to mask
>
> *Comment 5:* Input and output is not well defined. It's mentioned that it's an STFT, but it's not clear if it's power, magnitude, what's the STFT setup, and if any other transformation has been applied to it
>
> *********************************Response 5:********************************* We take STFT magnitude as the input. The STFT window is 50ms with 50% overlaps. No other preprocessing has been applied. We have published our code along with the dataset on GitHub [1].
>
> [1] IMUV dataset, 2022. URL [https://github.com/ICLR23-IMUV/IMUV](https://github.com/ICLR23-IMUV/IMUV).
>
> *Comment 6:* Section 3.2: “the testing data also follows the same training data distribution”. This makes Figure 7 relatively useless, since the average SI-SNR of the test data is expected to increase, i.e., denoising is becoming easier, so increasing SI-SNR is expected.
>
> ******************Response 6:****************** Note that as the number of clean samples increase, the self-supervised model gets more clean samples, and is more close to the supervised setting. Figure 7 is showing that IMU can provide consistent gain over audio-only denoiser. We have updated the text to emphasize the IMU gain.
>
> *Comment 7:* Reproducibility is definitely helped by the published GitHub repo. However, the paper is not clear enough on the parameter of the network and the training procedure.
>
> ******************Response 7:****************** We have published our code along with the dataset on GitHub [1]. Also we have publish the sample denoised audio on the anonymous GitHub site [2]
>
> [1] IMUV dataset, 2022. URL [https://github.com/ICLR23-IMUV/IMUV](https://github.com/ICLR23-IMUV/IMUV).
>
> [2] IMUV demo site, 2022. URL [https://iclr23-imuv.github.io/index.html](https://iclr23-imuv.github.io/index.html)

---

> > ### Comment · Reviewer_uCfP · 2022-12-01
> > **Thank you for your response**
> >
> > I appreciate responses from the authors.
> > However, I still have the same concerns related to this paper: the paper is relatively weak from both speech enhancement and ML perspective. My rating remains unchanged.

---

### Official Review · Reviewer_dV4m · 2022-10-24

**Confidence:** 4
**Correctness:** 3
**Technical Novelty And Significance:** 3
**Empirical Novelty And Significance:** 3
**Recommendation:** 5

**Clarity, Quality, Novelty And Reproducibility:**

Clarity

The paper is generally clear, but there are some missing details and clarifications that could be made (see weaknesses and minor comments above).

Quality

I think this is good quality paper. The quality could be improved by comparing to a stronger supervised baseline and addressing my comments in weaknesses.

Novelty

Conditioning separation on IMU has certainly been done before, but I think the proposed self-supervised approach is novel. It would be nice to have more discussion about why and how the proposed approach works. An ablation across correlation losses, and other relevant mechanisms, would be welcome.

Reproducibility

The paper releases the recorded data, which is great. And the github seems to have code to replicate training and evaluation. So reproducibility should be straight-forward.

**Strength And Weaknesses:**

Strengths

S1) The proposed self-supervised method seems effective at leveraging the additional conditional information provided by the IMU.

S2) The dataset (already released on anonymous github) seems to be the first publicly available dataset with audio and corresponding IMU.

Weaknesses

W1) The paper would benefit from some additional discussion of how the Translator can bootstrap its training by treating the (downsampled) noisy signal H as a target in the first stage. Does this work because the IMU is only correlated with the user's speech? Perhaps the correlation losses are quite important for this. An ablation with respect to the correlation losses would be interesting to see.

W2) It's not clear if (magnitude) spectrograms are used throughout the model. Is SI-SNR computed on waveforms, or spectrograms? If waveforms, how are the predicted spectrograms inverted back to waveforms? If spectrograms, it would be great to invert these back to waveforms and provide audio demos for listening.

W3) I think the supervised baseline could be stronger. It should be possible to directly train a single stage of the Denoiser part of the model pipeline, then the architecture would exactly match with the proposed method. For the existing baseline, please report comparison in terms of number of parameters and computation (e.g. FLOPs)

W4) I object to the use of the term "attention map" for the Translator's predicted M. This is just a mask, as it just elementwise multiplies a spectrogram. An attention map would be attending across values (i.e. multiplying and summing) given keys and queries. Please just call this a "mask".

W5) I have a number of minor comments on clarity, formatting, etc (see below)

Minor comments

m1) Spell out "SINR' acronym when first used, I think it's signal to interference and noise ratio.

m2) Maybe use italics instead of underlining in "Summary of results". Also, later on the format switched to using \texttt. Please pick a consistent format. i think italics throughout is fine.

m3) Some inconsistent capitalzation, e.g. "the Supervised audio Denoiser" -> "the supervised audio Ddenoiser"

m4) I find it a little odd to report percentage numbers

m5) "from the input data (H + L)": this should be "from the input data (H, L)", the + makes it seem like addition.

m6) "is the Correlation loss," -> "is the correlation loss,"

m7) In losses for Denoiser (by the way, would be good to add an equation number for these), the T(x) and D(x) notation strikes me as a little funny, but I guess it works.

m8) "Expectation-Maximization framework (Moon, 1996).": I think the most classic reference is Dempster, A.P.; Laird, N.M.; Rubin, D.B. (1977). "Maximum Likelihood from Incomplete Data via the EM Algorithm". Journal of the Royal Statistical Society, Series B. 39 (1): 1–38. JSTOR 2984875. MR 0501537.

m9) For the dataset description, please describe how much data in terms of duration was recorded. Also please report duration of the Google speech command dataset data that is used to construct the 990 hours of training data. It seems like the recorded data is not very big, did you observe issues with overfitting of the model? I generally find that having enough source data to create synthetic mixtures is important.

m10) This sentence seems unnecessary: "We call the target user Alice for ease of explanation."

m11) "distils" -> "distills"

m12) "we can either choose to obtain 1.5 to 4 dB SI-SNR gain": not clear which rows and columns this is comparing

m13) Do not italicize dB

m14) I think best practice is to use a single decimal place to report units in terms of decibels, since humans can barely even hear 0.1 dB of difference.

m15) Maybe swap order of Self-supervised IMUV and Supervised IMUV rows in Table 1?

m16) Spell out SIR when first used, I think it's signal to interference ratio

m17) "the SI-SNR drops if we remove the skip connection (0%).": what does 0% mean here?

m18) "and achieves promising Si-SDRi." -> "and achieves promising SI-SNRi"

m19) "operates entirely on 400 Hz, the viable sampling rate for real-world earphones. The energy budget for higher sampling rates substantively degrades the device’s battery-life.": Just because this reduces FLOPs? If the streaming network is small enough and/or the earphone accelerator is fast enough, it's also fine to process audio. But the paper does mention power consumption in the next sentence. Is that the main point of 400 Hz?

m20) "The IMU signal L is a projection of the audio signal H onto a lower dimensional space.": but isn't it also nonlinearly transformed?

**Summary Of The Paper:**

This paper proposes a self-supervised approach for training a speech enhancement network, given data from an inertial measurement unit (IMU) and audio recorded from an earphone microphone. The IMU captures a nonlinear and lower-frequency version of the user's speech through the bone condution channel. The proposed approach iterates between applying a "translator" model that predicts a mask M upsampled from the lower-frequency IMU signal L (uses 3 stages of upsampling, 400 Hz -> 800 Hz -> 3200 Hz -> 16 kHz). This mask M can be applied to the output of a "denoiser" stage, which takes audio mixture as input and conditions on the IMU signal L, and predicts denoised audio \hat{H}_T. Then the denoised audio is fed into another stage of the translator model, and the process is iterated. Experiments use 3 iterations of this process. The translator has a loss between the translator's output (the mask M applied to mixture audio H) and the denoiser output. The denoiser has multiple losses: reconstruction loss between output and translator's output, a loss that promotes part of its intermediate embedding at the middle of the U-Net matches L, and a correlation loss that minimizes the correlation between L and embedding for higher-level features (modeling the gap between audio and IMU), and maximizes correlation with the other component of Z and L.

The paper presents a new dataset of 4 participants with earphone and IMU, saying multiple keyword multiple times, and interfering speech audio is added. Performance is measured with SI-SNR (not clear if it's spectrogram or waveform domain) and keyword accuracy error. A supervised baseline with different architecture is trained on the same data. The proposed method is trained with noisy H (self-supervised) and clean speech for H (supervised). Method are also compared on personalized vs general setup. The proposed method improves in terms of SI-SNR, but is a bit worse in terms of keyword spotting error. Some ablation is done with respect to the skip connection in the U-Net.

Paper has two main contributions:
1) Self-supervised setup to learn from noisy audio with corresponding IMU
2) Dataset of audio recordings with corresponding IMU

**Summary Of The Review:**

Overall an interesting approach to learn to enhancement speech given an IMU signal without access to clean speech audio. The paper has some flaws that could be corrected (such as a stronger supervised baseline, ablations of the correlation losses, more clarity in presentation). If these weaknesses were addressed, I would be inclined to rate the paper higher. The paper releases data and code, which seems useful to the community.

---

> ### Author Response · Authors · 2022-11-19
> **Author response to reviewer dV4m**
>
> We thank the reviewer for all the insightful and constructive comments. We address some of the
> concerns below.
>
> *Comment 1:* How the Translator can bootstrap its training by treating the (downsampled) noisy signal H as a target in the first stage. Does this work because the IMU is only correlated with the user's speech?
>
> ***Response 1***: Yes. IMU only captures the jaw vibration from the target user, so the IMU signal is only correlated with the target speech. We add a paragraph describing the intuition at the end of the Translator Design in Sec. 2.2
>
>
>
> *Comment 2*: Is SI-SNR computed on waveforms, or spectrograms? If waveforms, how are the predicted spectrograms inverted back to waveforms? If spectrograms, it would be great to invert these back to waveforms and provide audio demos for listening.
>
> ************************Response 2:************************ We have added the audio demos of the denoised samples at the anonymous Github page:
> [https://iclr23-imuv.github.io/index.html](https://iclr23-imuv.github.io/index.html)
>
> We have updated the reference to this site in Sec. 3.1.
>
> *Comment 3:* The baseline denoiser should be possible to directly train a single stage of the Denoiser part of the model pipeline, then the architecture would exactly match with the proposed method. For the existing baseline, please report comparison in terms of number of parameters and computation.
>
> ************************Response 3:************************ We have performed new experiments with a new baseline supervised denoiser where the architecture exactly matches the proposed architecture. The SI-SNR results on User 1 in Figure 6 is 13.18 dB. Compared to the previous denoiser, the performance is better. Compared to the self-supervised IMUV (14.03dB), the supervised denoiser performance is worse without the help from IMU even if it is trained with the clean signal.
>
> We are reporting the number of parameters for all baseline models and proposed model.
>
> Supervised denoiser: 216K, supervised IMUV: 60K, self-supervised IMUV: 180K. We have updated these numbers in Sec. 3.1.
>
> *Comment 4:* I object to the use of the term "attention map" for the Translator's predicted M. This is just a mask, as it just elementwise multiplies a spectrogram.
>
> ************************Response 4:************************ We have renamed attention map to mask.
>
> *Comment 5:* Spell out "SINR' acronym when first used.
>
> ************************Response 5:************************ The correction has been made.
>
> *Minor comments* We accept most of the minor comments. Here we make inline responses to few comments below:
>
> *m9)* It seems like the recorded data is not very big, did you observe issues with overfitting of the
> model?
>
> We recorded ~1600 audio clips with IMU recordings in total. We mix these clips with the public Google speech dataset,  because (a) the noise speech distribution is drawn from a large public dataset (b) The words in the test set are different from words in the training set, we did not observe overfitting effect.
>
> *m19)* If the streaming network is small enough and/or the earphone accelerator is fast enough, it's also fine to process audio. But the paper does mention power consumption in the next sentence. Is that the main point of 400 Hz?
>
> Yes. Power consumption and computation overhead is the main concern. Normally earphone IMUs sample at a low rate to save power.  If high-sampling rate IMU is available, IMUV can be deployed without modification. Higher IMU sampling rate will only benefit our system.
>
> *m20)* “The IMU signal L is a projection of the audio signal H onto a lower dimensional space.": but isn't it also nonlinearly transformed?
>
> L is the aliased version of H filtered by the jaw channel. The aliasing function is linear addition, while the jaw channel might not be perfectly linear.

---

> > ### Comment · Reviewer_dV4m · 2022-11-30
> > **Thanks for your response**
> >
> > Thanks for the response, the authors addressed some of my concerns. Thank you for providing audio demos, they help me understand the task more.
> >
> > The authors did not provide information on how the spectrograms are inverted back to the time domain. I assume just an inverse STFT on masked input STFT?
> >
> > Also, after reading the other reviews and responses, I am still a bit concerned this paper doesn't compare to more state-of-the-art speech separation/enhancement baselines, as mentioned by other reviewers. Also, I have a couple concerns about the audio data and task itself, now that I understand it a bit better: the audio examples are quite short, and I wonder how this approach would work on longer utterances with the same setup. Also, I think that also considering non-speech interferers would improve the realism and applicability of the task. I noticed the response to reviewer NmkT, where the authors mentioned that UrbanSound8k was also injected, so I think these results should go into the paper as well. And some experiments should be done with non-speech-only interferers, as well as speech+non-speech interferers. In this respect too, as I look at the paper again, it often refers to "noises", but this is referring to speech interferers. I think this is a bit confusing, if the only interference is speech.
> >
> > I do think this is an interesting approach, but I think it's not quite over the bar for acceptance yet, and could benefit from some additional revision and experiments. Thus, I would like to keep my initial rating (5).

---

### Official Review · Reviewer_euGt · 2022-10-25

**Confidence:** 2
**Correctness:** 4
**Technical Novelty And Significance:** 3
**Empirical Novelty And Significance:** 3
**Recommendation:** 5

**Clarity, Quality, Novelty And Reproducibility:**

Clarity: very clear, except the concerns I listed under "Limitations". Figures are clear.
Quality and originality: The work seems original from many angles. Again consult "Limitations" for concerns.
Reproducibility: Authors have created publishable dataset. If code can be provided, it would be great. This is important since I think hyper-parameter choice is important here.

**Strength And Weaknesses:**

Strengths:
1. The experimental setup is vastly improved over the previous works.
2. In the world of earables, the choice of problem is very apt.

Limitations:
1. Perhaps a basic issue, but the intuition for why this should work is missing. It deserves atleast one paragraph.
2. Novelty should discussed further in Section 4 (Related work) under "Self-supervision".
3. Number of users to collect data is low. Age/gender details are missing.
4. Correlation loss under equation 2 has missing equation number. Also, why is weight not different for the two (postive and negative) terms?
5. Results could be more encouraging -- especially, given the fact supervised denoiser is working so well already.
6. Semi-supervised setup/fine-tuning may be explored in future.

**Summary Of The Paper:**

This paper aims at doing denoising (speech) without supervision. It pursues self-supervised learning while leveraging paired data from another modality (vibration IMU i.e. inertial measurement device). Speech is sampled at high rate while IMU data is sampled at only 400Hz. The proposed approach consists of two models (translator and denoiser) which are trained alternatively for 3 cycles. This optimization is seen akin to EM algorithm. The work is different from prior works in many respects. Previous works either don't explore self-supervised learning or use a more idealistic experimental setup. This work compares results with a regular Supervised Denoiser and find that when clean data is also available, their technique is close to Supervised Denoiser. When clean data is not available, the proposed approach helps a lot in SDR and KWS metrics.

**Summary Of The Review:**

The choice of problem is very apt given the current state of earable technology. It is interesting to see self-supervised paradigm being explored in an interesting setup. Denoising without clean data is a recently popularized problem which can have great implications in future. Authors develop connection to EM for their two-network alternating learning model. Results are encouraging but not very strong. Hence, this paper is of theoretical importance and is a good step in their choice of problem.

---

> ### Author Response · Authors · 2022-11-19
> **Author response to reviewer euGt**
>
> We thank the reviewer for all the insightful and constructive comments. We address some of the
> concerns below.
>
> *Comment 1*: the intuition for why this should work is missing
>
> ***Response 1***: We have added the intuition explanation in Sec. 2.2 at the end of both Translator Design and Denoiser Design
>
> *Comment 2*: Novelty should discussed further in Section 4 (Related work) under "Self-supervision”
>
> ***Response 2***: We have expanded the self-supervision section to describe the differences between IMUV and previous works.
>
> *Comment 3*: Correlation loss under equation 2 has missing equation number. Also, why is weight not different for the two (postive and negative) terms?
>
> ***Response 3***: We have fixed the equation number. For correlation loss, we tried to apply different weights for the two terms but it does not affect the results. This is because this loss is just used to provide a momentum to polarize the latent space (to separate the high frequency information for the target speech). As a result, the weight of the loss term does not matter much. Specifically, the gradient direction of these loss terms are similar.
>
> *Comment 4*: If code can be provided, it would be great
>
> ***Response 4***: We have added the code in the GitHub repository along with the dataset [1].
>
> [1] IMUV dataset, 2022. URL [https://github.com/ICLR23-IMUV/IMUV](https://github.com/ICLR23-IMUV/IMUV).

---

> > ### Comment · Reviewer_euGt · 2022-12-01
> > **Thanks for your response**
> >
> > I have slightly adjusted my scores. I think it is in the best interest if results can be improved further and concerns of other reviewers resolved in the next revision. I believe the technique carries great promise.

---

### Official Review · Reviewer_L1Xs · 2022-10-28

**Confidence:** 4
**Correctness:** 3
**Technical Novelty And Significance:** 1
**Empirical Novelty And Significance:** 2
**Recommendation:** 5

**Clarity, Quality, Novelty And Reproducibility:**

The technical depth and quality is quite minimal since the idea is mainly empirical.
The clarity of the paper can be improved. Reproducibility rests on whether these results can be replicated to larger datasets and with more complicated input features from the IMUs.

**Strength And Weaknesses:**

A meta comment is that, although the idea and proposal makes sense, since the evaluations set consisted of only 4 people, its uncertain how the proposal generalizes to large datasets, with realistic and noisy scenarios (e.g., noise in the collected samples).

1) Given that IMU signals are only relevant in low-frequency regimes, its important to understand the data more carefully here. Firstly how much variance there is in the IMU data that was collected from the 4 people; and if that is replicated to more people how would that change be?
2) We need to understand the trends as iterations progress i.e., how much quantifiably are the attention maps evolving as iterations increase and how much effect does that have on the overall performance?
3) Can you comment on the non-monotonic trends of self-supervised and supervised IMU models in Figure 8? Is this an aberration from data?
4) A natural baseline it to take the average of all the IMU inputs/spectra from the datasets and use that single average spectrum to regularize the enhancer? This is mainly because the IMU signals have smaller range compared to natural speech and so the networks may end up using an average signal from the dataset unless we have large variance in the dataset (goes back to comment 1) and meta comment above).
5) Speech enhancement community has been quite active in the past decade and there have been more audio-only enhancement algorithms since Park,Lee 2016. Whta was the rationale for not choosing more recent baselines?
6) Word recognition accuracy may not represent the subtle changes in signals resulting from different enhancement algorithms? What was the rationale for using this instead of say traditional perceptual/quality metrics like pesq, mos etc.

**Summary Of The Paper:**

The authors propose to utilize IMU signals for providing self supervision to audio signals for the problem of speech enhancement. This in itself is a good idea, and some that reasonable from the perspective that bone conduction is measurable for low level frequencies. The authors propose a simple algorithm the evaluate this. The evaluations are minimal and the presentation is good.

**Summary Of The Review:**

As mentioned in the weaknesses, the main aspect is generalizability to reasonable sized datasets. The idea in itself is good/interesting. Additionally many details about the data, the model and its robustness are missing, and will improve the contribution.

---

> ### Author Response · Authors · 2022-11-19
> **Author response to reviewer L1Xs**
>
> We thank the reviewer for all the insightful and constructive comments. We address some of the
> concerns below.
>
> *Comment 1*: How much variance there is in the IMU data that was collected from the 4 people; and if that is replicated to more people how would that change be?
>
> ***Response 1***: Because each person has a unique bone channel, the IMU recordings will be quite different across different users speaking the same word. We have published the collected IMU data on the Github [1]. Since we are training a personal model for each user (e.g., the earphone will train a personal model for each user), the IMU variance across users will not increase the model complexity.
>
> [1] IMUV dataset, 2022. URL [https://github.com/ICLR23-IMUV/IMUV](https://github.com/ICLR23-IMUV/IMUV).
>
> *Comment 2: We need to understand the trends as iterations progress i.e., how much quantifiably are the attention maps evolving as iterations increase and how much effect does that have on the overall performance?*
>
> ***Response 2***: We do not show the mask progress in the paper for the interest of space. However, we have uploaded mask images as samples to the GitHub repository, the samples can be found in [2].
>
> As iteration progresses, the  Denoiser will confirm the correctness of the mask, so the mask will get sharper and remove more noise. Specifically, the mask in the first iteration just suppresses the noise rather than removing it. In the following iteration, the Denoiser then confirms the correlation between the mask and the noisy audio, and the Translator will output a sharp mask accordingly. In most cases, the mask will converge in two cycles.
>
> [2] IMUV sample output for Translator and Denoiser. URL [https://github.com/ICLR23-IMUV/IMUV/tree/main/Examples/Figures](https://github.com/ICLR23-IMUV/IMUV/tree/main/Examples/Figures)
>
> *Comment 3*: Comment on the non-monotonic trends of self-supervised and supervised IMU models in Figure 8.
>
> ***Response 3***: The gain of self-supervised IMUV over supervised Denoiser does not increase monotonically. In other words, self-supervised IMUV performs better in the 5 dB scenario. The reason is as follows:
>
> At 0 dB SINR, the self-supervised scheme can be penalized if the mask includes some noise in the T-F bins. With strong noises, the Translator output will contain comparable noises in the clean signal approximate. On the other hand, when the SINR goes high, even if the mask selects some of the T-F bins from the interference, it will not be translated to large noise amplitude in the Denoiser loss function. For 10 dB SINR case, it’s hard for the self-supervised scheme to suppress the small noises without clean speech input. In other words, it’ performance saturates at ~15dB, which is ideal for ASR applications.
>
> *Comment 4:* A natural baseline it to take the average of all the IMU inputs/spectra from the datasets and use that single average spectrum to regularize the enhancer?
>
> ***Response 4:*** Because IMU is the aliased target speech signal, the IMU recordings are very different across different words and user. Averaging the IMU spectra will not give information about the target speech of different words. As shown in Figure 1 and Figure 4, the IMU spectrogram as well as the translated mask clearly indicate (a) the base frequency and (b) the occurrence time of the target speech.
>
> *Comment 5*: What was the rationale for not choosing more recent baselines?
>
> ***Response 5:***  There are two main reasons for the selection:
>
> (a) Because we are targeting the earphone application, we select a flexible learning model that can converge quickly on smaller personal dataset. Recent audio-only works such as pseudo-SE [1] or MixIT [2] require training on large public audio dataset.
>
> (b) Our model aims to denoise the human speech interference. Previous audio-only denoisers optimize for ambient noise dataset like FreeSound [3], which has a very different distribution. As a result, we select a small network that can converge on the speech interference quickly.
>
> [1] Sivaraman, A., Kim, S., & Kim, M. (2021). Personalized speech enhancement through self-supervised data augmentation and purification. *arXiv preprint arXiv:2104.02018*
>
> [2] Wisdom, S., Tzinis, E., Erdogan, H., Weiss, R., Wilson, K., & Hershey, J. (2020). Unsupervised sound separation using mixture invariant training. *Advances in Neural Information Processing Systems*, *33*, 3846-3857.
>
> [3] Freesound. (n.d.). Retrieved November 18, 2022. URL: [https://freesound.org/](https://freesound.org/)

---

### Official Review · Reviewer_NmkT · 2022-11-03

**Confidence:** 4
**Correctness:** 3
**Technical Novelty And Significance:** 3
**Empirical Novelty And Significance:** 3
**Recommendation:** 5

**Clarity, Quality, Novelty And Reproducibility:**

The paper is a bit hard to follow due to changing math notation. More consistent notation would help. For example, the variable x in equations after Equation (2) is not consistent with earlier notation. T and D networks have different inputs, so using x for both is confusing. The training algorithm is confusing to describe, so making the code available would help for reproducibility.

**Details Of Ethics Concerns:**

No ethics concerns.

**Strength And Weaknesses:**

Strengths:
1. New speech + IMU dataset.
2. New self-supervised enhancement model based on alternating minimization training.

Weaknesses:
1. Comparison with more direct self-supervised techniques like pseudo-SE or Mixit like training when adding IMU data to the input.
2. Dataset assumes there is always interfering speech which is actually quite rate in real use.

**Summary Of The Paper:**

A self-supervised speech enhancement method that uses both audio and IMU data is presented. The method is self-supervised in the sense that the audio does not need to be a clean audio signal. The method is targeted towards earphone recorded audio data.

The self-supervision is handled with an alternating minimization like training algorithm where a "translator" model that maps IMU recordings to spectrogram masks that multiply the input audio spectrogram and another "denoiser" model that takes both IMU and input spectrograms and tries to predict a clean spectrogram without having access to the clean spectrogram. The denoiser model has an internal representation of the input audio and IMU and it tries to pick the part of the representation that correlates with the IMU data to reconstruct the spectrogram. By doing this, it is assumed that the part of the audio that correlates with the IMU is estimated only which refines the estimated clean audio output. This new estimate is used in the "translator" loss function once again. Thus, each module provides the target for the other module in an alternating fashion. Additional "correlation" losses are used in the denoiser to force the model to output signals that correlate with IMU data.

**Summary Of The Review:**

The paper introduces a method for self-supervised training of speech enhancement models.

I think the method described in the paper has a risk of divergence during training due to complexity of the training method. I think there are more straightforward ways to perform self-supervised training for speech enhancement. One of them is to use MixIT method for speech enhancement that is described in the MixIT paper. In that approach, we add noises to existing data (whether the data is clean or not) and then separate it into three outputs which should satisfy the MixIT loss. A simplification of this method is something called pseudo-speech-enhancement in a paper (Siwaraman, Kim and Kim) which can also be tried. In that one, we just add noise to data and assume the data is clean whether it is clean or not.

In the dataset, an interfering speech is added to the clean speech utterances but it is not always realistic (as also mentioned in the paper). The interference could be environmental noise, or silence or another speaker. It would be more interesting to study the case where each alternative is added with some probability. Also, the interference would typically have reverberation which is not used in the dataset.

In the models for comparison, it is not clear how "Supervised IMUV" is trained. Is it trained using alternating minimization, or is it trained in a straightforward way with noisy input and clean output (so no translator is used?) ? This should be clarified. Also, for "supervised denoiser" model, why not use the same architecture without the IMU input?

In Table 1, when using a general model, the Si-SNR gets worse for the "supervised denoiser" which probably means the data is not enough to train a reliable denoiser.

In the dataset, the interference is at a fixed SIR. In Figure 8, the SIR value is changed, but it is the same for train and test data. I think to be more realistic, the SIR values should be in a realistic range for both training and test data.

---

> ### Author Response · Authors · 2022-11-19
> **Author response to Reviewer NmkT**
>
> We thank the reviewer for all the insightful and constructive comments. We address some of the
> concerns below.
>
> *Comment 1*: Comparison with more direct self-supervised techniques like pseudo-SE or MixIT-like training when adding IMU data to the input.
>
> ***Response 1***: We thank the reviewer for pointing us to the helpful related work. For pseudo-SE, it targets removal of none-speech interference by adding known noises to the recorded-premixtures. In our scenario, if the interference comes from another speaker, the system might require a larger amount of data to converge.
>
> For MixIT, it presents an excellent gain in both speech separation and speech enhancement. Self-supervised IMUV gain in 0 dB SINR setting (the source separation scenario) is 8.21 dB, which aligns with the results of MixIT source separation gain (~10 dB on anechoic setting and 4 dB in reverberant setting).
>
> However, it does not support speech enhancement where the interference is also a speech. For speech separation, it assumes the power of two speeches to be comparable. This does not align with the earphone usecase, where the target speech will have higher SINR.
>
> *Comment 2*: I think the method described in the paper has a risk of divergence during training due to complexity of the training method.
>
> ***Response 2***:  In IMUV case, because the Translator outputs a mask that only removes components from the original noisy signal, the approximate clean signal, i.e., the latent distribution in EM, will not exceed the original input.  In other words, the worst case is that the Denoiser gets the noisy input as a target from the Translator, which will not diverge.
>
> The convergence of Expectation-Maximization methods greatly depend on the initial value. Because IMU is the aliased version of the clean audio, the Translator can generate a good mask as a initial value. Empirically, the mask converges within two cycles most of the time.
>
> *Comment 3*: The interference could be environmental noise, or silence or another speaker.
> It would be more interesting to study the case where each alternative is added with some probability
>
> ***Response 3***: We also added UrbanSound8K [1] as the environmental noise in addition to the interferer. We did not report this benchmark result because of the interest of space.  The results for SINR=5 dB are as follows: supervised IMUV: 11.92 dB, self-supervised IMUV: 8.37 dB. Accordingly, IMUV also works well with the presence of environmental noise. We also evaluate the silence cases. In Figure 7, we inject different percentages of clean audios in the training and testing set.
>
> [1] J. Salamon, C. Jacoby, and J. P. Bello, “A dataset and taxonomy for urban sound research,” in
> 375 Proceedings of the 22nd ACM international conference on Multimedia, 2014, pp. 1041–1044.
>
> *Comment 4*: In the models for comparison, it is not clear how "Supervised IMUV" is trained.
>
> ***Response 4:*** It is trained with the noisy input and clean output directly. With the clean speech available, we don’t need to run the Translator to get a clean speech approximate.
>
> *Comment 5*: when using a general model, the Si-SNR gets worse for the "supervised denoiser" which probably means the data is not enough to train a reliable denoiser.
>
> ***Response 5***: In the general model setting, the training data does not contain the speech from the target user. As a result, the supervised denoiser has no clue about who the target is. If the denoiser cannot identify the target user, it might cancel the target user speech so the results get worse.
>
> We introduce the general model setting because this is the challenge where a new earphone does not have the speech from the target user. On the contrary, IMU can help identify the target user signal (even if it’s not present in the training dataset), so IMUV can still present some gain over the raw mixtures in this case.
>
> *Comment 6*:  In Figure 8, the SIR value is changed, but it is the same for train and test data. I think to be more realistic, the SIR values should be in a realistic range for both training and test data
>
> ***Response 6:*** In Figure 8, we want to prove that IMUV works under different SIR regimes. Theoretically, since we do not use SIR as our objective function, the mismatch SIR between the training and testing dataset will not affect the results.

---

> > ### Comment · Reviewer_NmkT · 2022-12-02
> > **Post-rebuttal**
> >
> > Thanks to the authors for their response and clarifications.
> >
> > Now, I see that a supervised denoiser for a general model does not work because the interference is another speaker. However, if we know that the interference is another speech, we should run a speech separation model instead of a denoiser model for a fairer comparison.
> >
> > A better dataset collection with various interference types and comparison with existing self-supervised methods would help improve the paper. MixIT can be applied to interfering speech case too with some small changes to the speech enhancement use case proposal. MixIT was used for speech separation in the original paper. It can also be used when we do not know whether the interference is non-speech or speech. It may not perform as well in those cases, but that remains to be seen especially with IMU input, it may learn to make use of the information to improve its performance.
> >
> > Using the same SIR for interference for both train and test data is a matching condition. We would expect our models work in a range of conditions, not for a fixed condition. Even when SIR is not used in the loss function, the train and test match or mismatch would affect the performance and it is important to report the performance in more realistic conditions rather than in matching conditions only.
> >
> > Based on these remaining concerns, I decide to keep my original score.

---

### Decision · Program_Chairs · 2023-01-20

**Decision:**

Reject

**Justification For Why Not Higher Score:**

Although the technique is interesting, evaluations and presentation are both lacking. The novelty is not enough to overcome these limitations.

**Justification For Why Not Lower Score:**

N/A

**Metareview: Summary, Strengths And Weaknesses:**

The authors present a technique for self-supervised learning using audio and IMU data. Targeting earphones / earables, wherein such information is available, the presented technique has the advantage that one does not need access to ground truth clean data for training. Broadly, the approach works by finding correlations between noisy audio and IMU data, which is not affected by background noise.

The reviewers pointed out that there is novelty in how the authors use the 2 modalities for optimization in an alternating fashion, without the need for clean data. The authors also open source a new dataset for supporting more work in this space, which is great.

Multiple reviewers pointed out shortcomings with evaluations. For example, comparisons with other more direct self-supervised techniques using IMU data as an additional modality would have been useful, along with other more recent approaches. Importantly, the training and test settings are also not diverse enough, making it challenging to assess the overall quality. The reviewers also brought up clarity of presentation as a weak point of the submission.

**Summary Of Ac-Reviewer Meeting:**

N/A